# BAYESIAN OPTIMAL EXPERIMENTAL DESIGN FOR THE SURVEY BANDIT SETTING

## ABSTRACT

The contextual bandit is a classic problem in sequential decision making under uncertainty that finds broad application to tasks in precision medicine, personalized education, and drug discovery. Here, a decision maker repeatedly receives a context, takes an action, and then observes an associated outcome, with the goal of choosing actions that achieve a minimal regret. However, in many settings, the context is not given, and the decision maker must instead collect some information to infer a context before proceeding. For example, when a doctor does not have prior information about a patient, they might ask a sequence of questions before recommending a medical treatment. In this paper, we aim to develop methods for this setting—which we refer to as the *survey bandit*—where the decision maker is not given access to the context but can ask a finite sequence of questions to gain information about the context before taking an action and observing an outcome. Using insights from Bayesian optimal experimental design (BOED) and decision-theoretic information theory, we view the interaction with each user as a BOED task, where the goal is to ask a sequence of questions that elicit the most information about the optimal action for this user. Our procedure is agnostic to the choice of probabilistic model, and we demonstrate its usefulness in a few common classes of distributions. Our algorithm achieves significantly better performance on both synthetic and real data relative to existing baseline methods while remaining statistically efficient, interpretable, and computationally friendly.

## 1 INTRODUCTION

In many sequential decision making applications, a decision maker faces a sequence of users, for which they need to choose an action and then observe an outcome. Each user has a context vector (i.e. a set of features), which, in many cases, is not known a priori to the decision maker[1]. The context is needed to choose an action that yields a good outcome, but acquiring this context can be expensive or time-consuming. We refer to this setting as the *survey bandit*, which has been previously studied by Krishnamurthy & Athey (2020). One example of this setting is in personalized medicine: a physician faces a sequence of patients, and to each they can ask a few questions before recommending a final treatment (Yao et al., 2021; Tomkins et al., 2021). Another example can be found in education: during office hours, a professor faces a sequence of students, and they can ask each a few questions before recommending an exercise or a reading. Last but not least, the survey bandit setting also finds application in drug and material discovery: during virtual screening, a chemist faces a large set of molecular structures, and they can perform a finite set of tests on each candidate (e.g. DFT calculation or molecular docking) before deciding whether or not it should go on to the next phase of the study (Kitchen et al., 2004; Bengio et al., 2021; Svensson et al., 2022). Users' context features are usually not independent, and thus good decision making can be achieved even when a small part of the context is observed. For example, in a series of questions related to political leaning, if the decision maker observes that a user prefers to watch Fox News, they may not need to ask whether or not they identify as a conservative.

Suppose the decision maker can sequentially ask a few questions before recommending a treatment—what questions should they ask? One way to tackle this problem is to view querying

---

[1]This setting also includes cases where the decision maker has partial information, such as a prior belief about the users' contexts, before asking questions.

answers from the user as a feature selection problem, where only a small subset of features is useful to predict the outcome (Bastani & Bayati, 2020). Taking this view, Krishnamurthy & Athey (2020) treat querying answers as feature selection using ridge regression. Using the linear payoff assumption, RidgeUCB further assumes the knowledge of a threshold $\beta_{\min}$ such that features with a ridge regression coefficient below this threshold have no impact on the outcome, and hence can be ignored. Although this assumption is intuitive, RidgeUCB can be brittle when the assumption is violated. In practice, it is unclear how to set $\beta_{\min}$ without knowing the strength of the relationship between contexts and outcomes. Taking a similar perspective, Bouneffouf et al. (2017) views the question phase in the survey bandit setting as a feature selection problem and proposes the Contextual Bandit with Restricted Context algorithm as a solution. The feature selection view of the survey bandit can additionally introduce a challenging combinatorial search problem (i.e. in choosing an optimal subset of the features).

This paper takes an alternative point of view. We exploit the ability to sequentially query features and receive a signal from the user in the survey phase to adaptively ask the most informative question, following insights from Bayesian optimal experimental design (BOED) (Chaloner & Verdinelli, 1995; Ryan et al., 2016) and decision-theoretic information theory (DeGroot, 1962; Rao, 1984; Neiswanger et al., 2022). This alternative approach treats the question phase for each user as a BOED problem, where the goal is to ask the most informative question, while the treatment phase can be formulated as a contextual bandit problem. Instead of eliminating the feature that is believed to be unimportant, our approach tries to model the dependencies between features by leveraging probabilistic modeling and approximate inference, in order to carry out a sequence of decision making tasks. To the best of our knowledge, this hybrid approach between BOED and contextual bandits has not yet been explored for the survey bandit setting.

In full, our method takes advantage of a recently-developed decision-theoretic BOED approach, which allows us to identify the question that elicits the most information about the best action for a given user in expectation. We conduct experiments on synthetic and real datasets in the survey bandit setting and show strong performance relative to a number of baselines. Our method has intimate connections with a variety of algorithms for decision-making under uncertainty, such as Bayesian optimization, active learning, and contextual bandits. All implementations will be made publicly available.

## 2 DECISION-THEORETIC ENTROPY SEARCH FOR SURVEY BANDIT

We start by establishing some notation for the survey bandit setting. Facing a sequence of $U$ total users, a decision maker can ask each user a fixed number of questions, observe their answers, recommend a treatment, and observe a corresponding outcome. We assume there are a total of $Q$ questions and $T$ treatments that a decision maker can choose from. For each user, the number of questions that the decision maker can ask, denoted $Q_{\text{allow}}$, is assumed to be given. Associated with each user is a vector $y \sim p(y), y \in \mathbb{R}^{Q+T}$, which can be partitioned into an answer vector $y_a = [y_{a,1}, ..., y_{a,Q}]$ and an outcome vector $y_o = [y_{o,1}, ..., y_{o,T}]$. We can also partition $y$ into an observed vector $y_{\text{obs}}$ and an unobserved vector $y_{\text{unobs}}$. The decision maker's goal is to ask, for each user, informative questions that reveal the treatment with the highest outcome, in expectation. A graphical illustration of the problem setting is given in Figure 1.

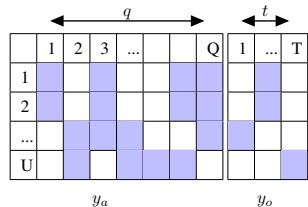

Figure 1: Each row of the above matrix is a user. The shaded and unshaded cells are observed and unobserved answers/outcomes, respectively. Here, $Q_{\text{allow}} = 4$ is shown.

### 2.1 DECISION-THEORETIC ENTROPY SEARCH

In this subsection, we assume that the joint distribution $p(y_a, y_o)$ is given, and in the following section we will discuss how to estimate this distribution from observed data. Given this joint density, the decision maker should ask the question that elicits the most information—i.e. the greatest reduction in posterior uncertainty—about the best treatment, on average. One notion of mutual information between an answer and the best outcome can be captured by $H_{r,\mathcal{A}}$-entropy, a decision-theoretic notion of uncertainty (DeGroot, 1962; Rao, 1984; Neiswanger et al., 2022). Suppose we have a Bayesian model for a parameter $\phi \in \Phi$. Then the $H_{r,\mathcal{A}}$-entropy is parameterized by a prior distribution $p(\phi)$,

---

**Algorithm 1** $H_{r,\mathcal{A}}$-Entropy Search (HES) for the Survey Bandit Setting

---

$\mathcal{D} \leftarrow \emptyset$
**for** $u \leftarrow 1$ to $U$ **do**                                           ▷ Iterate through each user $u \in [U]$
    $\theta_u \leftarrow$ Estimate model parameters using $\mathcal{D}$
    $\mathcal{D}_u \leftarrow \emptyset$
    **for** $q \leftarrow 1$ to $Q_{\text{allow}}$ **do**
        **for** $q' \leftarrow 1$ to $Q$ **do**                        ▷ Iterate through each question $q' \in [Q]$
            $\{y_{a,q'}^i\}_{i=1}^I \overset{\text{iid}}{\sim} p(y_{a,q'}|\mathcal{D}_u; \theta_u)$           ▷ Draw samples for the MC estimate
            **for** $i \leftarrow 1$ to $I$ **do**
                Compute $\mu(y_o|y_{a,q'}^i, \mathcal{D}_u; \theta_u)$         ▷ Compute conditional distribution
            **end for**
        **end for**
        $q^* \leftarrow \arg\max_{q' \in [Q]} \text{EHIG}(q')$           ▷ Optimize EHIG acquisition function
        $\mathcal{D}_u \leftarrow \mathcal{D}_u \cup \{y_{a,q^*}\}$           ▷ Update user dataset with observed answer
    **end for**
    **if** $\epsilon > z \sim U(0,1)$ **then**           ▷ Follow a decaying $\epsilon$-greedy strategy
        $t^* \leftarrow$ Select treatment that maximizes $\mu(y_{o,t}|\mathcal{D})$
    **else**
        $t^* \leftarrow$ Select treatment as draw from $\text{Unif}([T])$
    **end if**
    $\mathcal{D}_u \leftarrow \mathcal{D}_u \cup \{y_{o,t^*}\}$           ▷ Update user dataset with observed outcome
    $\mathcal{D} \leftarrow \mathcal{D} \cup \{\mathcal{D}_u\}, \quad \epsilon \leftarrow \epsilon \times \alpha$          ▷ Update full dataset and $\epsilon$
**end for**

---

and reward function $r : \Phi \times \mathcal{A} \to \mathbb{R}$. Given a dataset $\mathcal{D}$, the posterior $H_{r,\mathcal{A}}$-entropy is defined as $H_{r,\mathcal{A}}[p(\phi|\mathcal{D})] = -\sup_{a \in \mathcal{A}} \mathbb{E}_{p(\phi|\mathcal{D})}[r(\phi, a)]$.

For the survey bandit setting, the unknown vector of interest is the outcome vector $\phi = y_o$ and we will be interested in the uncertainty (entropy) of the posterior distribution over this outcome vector, $H_{r,\mathcal{A}}[p(y_o|\mathcal{D})]$. Here, the action set is the set of treatment indices, i.e. $\mathcal{A} = \{1, ..., T\} = [T]$, and the dataset $\mathcal{D}$ contains the answers to a sequence of questions asked to a given user. In this setting, the reward function evaluated at a treatment $t$ and outcome vector $y_o$ is defined to be $r(y_o, t) = y_{o,t}$, i.e. the reward function indexes into the vector $y_o$ at position $t$. For a given user, the myopic Bayesian optimal question, which has the greatest expected increase in information about the optimal treatment, is the one returned by maximizing the expected $H_{r,\mathcal{A}}$-information gain (EHIG) (Neiswanger et al., 2022):

$$\underset{q \in [Q]}{\arg\max} \text{EHIG}(q; r, \mathcal{A}) = \underset{q \in [Q]}{\arg\max} \left( H_{r,\mathcal{A}}[p(y_o|\mathcal{D})] - \mathbb{E}_{p(y_{a,q}|\mathcal{D})} \left[ H_{r,\mathcal{A}}[p(y_o|y_{a,q}, \mathcal{D})] \right] \right)$$

$$= \underset{q \in [Q]}{\arg\max} \mathbb{E}_{p(y_{a,q}|\mathcal{D})} \left[ \max_{t \in [T]} \mathbb{E}_{p(y_o|y_{a,q}, \mathcal{D})}[r(y_o, t)] \right] \tag{1}$$

$$\approx \underset{q \in [Q]}{\arg\max} \frac{1}{I} \sum_i^I \left[ \max_{t \in [T]} \mu(y_{o,t}|y_{a,q}^i, \mathcal{D}) \right],$$

where the final expression is a Monte Carlo estimate given by $I$ samples $\{y_{a,q}^i\}_{i-1}^I \overset{\text{iid}}{\sim} p(y_{a,q}|\mathcal{D})$ drawn from the posterior distribution over answers to question $q$. Note that, to compute the expectation over $p(y_o|y_{a,q}, \mathcal{D})$ in the final expression, the EHIG only needs the mean $\mu(y_o|y_{a,q}, \mathcal{D})$. The two discrete optimizations are carried out by enumerating over sets $[Q]$ and $[T]$. Both Monte Carlo approximation and optimization over questions and treatments can be efficiently parallelized on modern computers with vectorization.

Choosing questions according to Equation (1) is an example of the $H_{r,\mathcal{A}}$-entropy search (HES) algorithm. The full HES procedure for the survey bandit setting is given in Algorithm 1. After the question phrase, HES recommends a treatment that maximizes the conditional expected outcome. This treatment recommendation can be viewed as a greedy (pure exploitation) action from a contextual bandit perspective, and following the estimated optimal action only will lead to suboptimal exploration. To encourage exploration, a decaying $\epsilon$-greedy strategy (Auer et al., 2002) is employed,

which is a simple yet often effective strategy. Other strategies, such as UCB, could be used as well to balance exploration and exploitation. Note that, in Algorithm 1, computing the conditional distribution and estimating parameters are repeated many times, which imposes additional computational constraints on the choice of the probabilistic model over answers and outcomes.

## 2.2 THE JOINT DISTRIBUTION OF USERS' ANSWERS AND OUTCOMES

In this section, we discuss models for the joint distribution $p(y_a, y_o; \theta)$ that capture dependencies between answer and outcome vectors. As stated previously, the family $p(y_a, y_o; \theta)$ needs to allow conditioning efficiently and parameter estimation with missing data. Both need to be done in a computationally efficient manner since the HES procedure iteratively performs these steps as each new user arrives. A good class of parametric distributions that satisfies these criteria is the Gaussian mixture model (GMM) (Murphy, 2012). A GMM can capture the complex geometry of many real-world distributions while allowing closed-form conditioning and efficient parameter estimation. While we focus on GMMs as demonstrations in this paper, extensions to other probabilistic models are straightforward. Below, we briefly review GMMs and the procedures to estimate their parameters under missing data, which we will require for our method.

A GMM generates data by the following hierarchical process:

$$c \sim \text{Mutinomial}(c; 1, \pi), \qquad y_a, y_o \sim \mathcal{N}(y_a, y_o; \mu^c, \Sigma^c) \tag{2}$$

where $c$ is a one-hot vector that denotes the cluster to which the generated sample $y$ belongs. $c$ is drawn from a single trial multinomial distribution with cluster proportion $\pi = [\pi_1, ..., \pi_C], \sum_{c=1}^{C} \pi_c = 1$. In Figure 2, we show a graphical model in plate notation for the above generative process, which we extend to include $U$ users and potentially noisy observations of answers and outcomes, $y_{\text{obs}}$. In contrast with a typical GMM, our model for the survey bandit setting involves an additional set of observed variables, whose elements might vary from user to user.

The conditional mean, covariance, and proportion of each mixture model component can be updated with Bayes' rule or in closed form. We note that the EHIG optimization procedure (in Algorithm 1) only requires the following three quantities:

1. The distribution of a user's cluster assignment, conditioned on their observed answers, $p(c|y_{a,\text{obs}})$.

2. The distribution of a user's unobserved answers, conditioned on their observed answers, $p(y_{a,\text{unobs}}|y_{a,\text{obs}})$.

3. The mean of a user's outcome distribution, conditioned on their observed answers, $\mu(y_o|y_{a,\text{obs}})$.

All three of these quantities can be estimated (see Appendix A for detailed formula), via the following quantities:

1. The empirical mean of outcome and answer vectors from a sample of users, $\mu(y_o)$ and $\mu(y_a)$.

2. The empirical covariance between answers from a sample of users, $\Sigma_{aa}$.

3. The empirical covariance between answers and outcomes from a sample of users, $\Sigma_{ao}$.

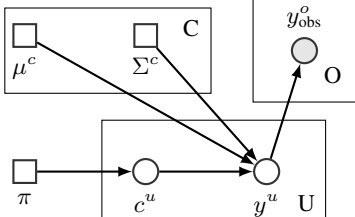

Figure 2: GMM graphical model in the survey bandit setting. Shaded and unshaded nodes denote observed (potentially with noise) and latent variables, respectively. The observed component of $y$ is different for each user. $C$, $U$, and $O$ denote the total number of clusters, the total number of users, and the total number of observed questions and outcomes, respectively.

The covariance between outcomes, $\Sigma_{oo}$, is impossible to estimate without further assumptions due to the Fundamental Problem of Causal Inference (Holland, 1986), where no two outcomes can be observed simultaneously. Fortunately, the EHIG optimization expression in Equation (1) does not require this quantity.

GMM parameters are estimated in the presence of missing data with the *expectation-conditional maximization (ECM)* algorithm (Meng & Rubin, 1993; Ghahramani & Jordan, 1993; McCaw et al., 2022). ECM is a generalization of the *expectation-maximization (EM)* algorithm (Dempster et al., 1977), a popular algorithm for maximum likelihood estimation (MLE) in GMMs. ECM replaces the

maximization step over all parameters of interest in EM with a sequence of conditional maximization steps, where each parameter of interest is optimized individually conditioned on other parameters. ECM requires missingness in the input to satisfy the *missing at random (MAR)* assumption for unbiased estimation, meaning whether an element in the input vector is missing can depend on the observed elements, but not on the unobserved elements. This requirement is satisfied in our setting, because whether or not HES chooses to observe a context does not depend on the unobserved data. For a detailed treatment of ECM, we refer the reader to Meng & Rubin (1993); Dempster et al. (1977); Ghahramani & Jordan (1993), and (McCaw et al., 2022). MLE is used in this study because it is computationally efficient, but other inference methods, such as Markov chain Monte Carlo or variational inference, can be used as well.

The dataset in the survey bandit setting has a large amount of missing data for two reasons: (1) the context for each user is only observed partially and at a low rate, and (2) only one treatment per user is observed. This large amount of missing data can affect the stability and convergence rate of the iterative optimization procedure in ECM (McCaw et al., 2022). In Appendix A, we describe our initialization procedure to stabilize ECM in the presence of missing data.

## 3 EXPERIMENTS

We first verify that HES works as advertised on six artificial data generating processes, where details about the set of informative questions and treatments are known. We then conduct experiments on a real dataset from the charitable giving experiment (Athey et al., 2022). Each algorithm is evaluated with four metrics: (1) cumulative regret, (2) per-period regret, (3) per-period accuracy of estimating the optimal set of questions/treatments for the population, and (4) per-period accuracy (measured by MAE) of the outcome model. Regret is defined as the difference between the outcome of treatment chosen by the algorithm and the optimal outcome. Per-period regret and per-period accuracy (of question, treatment, and outcome estimation) are computed every 100 users. For the per-period accuracy of estimating the optimal set of questions/treatments for the population, we compute the fraction of the *set* of optimal questions/treatments that the algorithm correctly identifies[2]. Additional details on the experimental setup are given in Appendix B. HES is compared against the following six baseline algorithms.

1. LinUCB (Li et al., 2010) with full access to answers: a standard contextual bandit algorithm *with access to the ground truth context*, which represents what can be achieved when there is no budget constraint on question selection.

2. RidgeUCB (Krishnamurthy & Athey, 2020): as discussed previously in Section 1, RidgeUCB treats question selection as feature selection, where features with sufficiently high coefficients in a ridge regression are selected, and then uses LinUCB to recommend treatment.

3. Uncertainty Sampling (US) (Lewis & Gale, 1994), a popular strategy in active learning: this uses a probabilistic model of answers and outcomes to query questions with the highest predictive uncertainty, and then selects treatments with the highest expected conditional outcome.

4. Uniform Non-contextual Randomization (UNR) (Hariton & Locascio, 2018), a popular method for conducting non-contextual experiments: this recommends treatments uniformly at random without considering the answers to questions. UNR is expected to have constant per-period regret over time, but is useful as a benchmark for how data collection influences the quality of learned models of the answer distribution and the outcomes conditional on answers.

5. UniformBOED (Fang & Lin, 2003): this method selects questions to query uniformly at random, and then selects the treatments with the highest expected conditional outcome according to a probabilistic model.

6. BayesLinUCB: this method queries questions uniformly at random, infers the full context by a probabilistic model, and then uses LinUCB to recommend treatments (Levine, 2018).

---

[2]This is not necessarily the per-user optimal questions/treatment. For example, if a population has informative questions as 1, 2, and 3 and $Q_{\text{allow}} = 3$, and the algorithm asks 2, 3, 4, its accuracy of identifying the set of optimal questions is $\frac{2}{3}$. If the set of optimal treatments for the population is $\{1, 2, 3\}$, and the algorithm chooses treatment 2 for a user, its accuracy of selecting the optimal set of treatments is 1, even if the true optimal treatment for this user might not be 2.

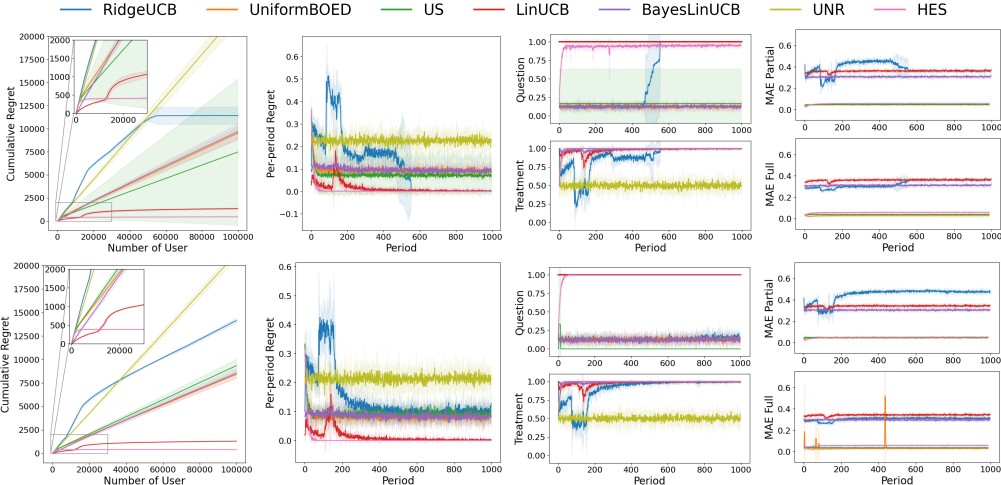

Figure 3: *A Simple Homogeneous Population* (top) and *A Homogeneous Population with Answer Dependency* (bottom). Summary statistics of all metrics are given in Table 1 and 2 in Appendix B.

Each simulated experiment is run for a sequence of 100,000 users. In the beginning, each algorithm has a warm-up period where it can interact with 30 users per treatment. During this period, the decision maker can ask users all possible questions and recommend one treatment per user. There are a total of 16 questions and 6 treatments in all experiments. The parameters of the GMM are updated every 500 users. Parameters of LinUCB and ridge regression are updated with every user. RidgeUCB is modified slightly to enable it to handle budget constraints in the number of questions: the algorithm is allowed to observe up to a size $Q_{\text{allow}}$ questions, which are randomly drawn from the set of questions that it requests to ask[3]. Empirically, this modification does not change RidgeUCB behavior significantly because the algorithm rarely exceeds the budget constraint with the default $\beta_{\min} = 0.5$. Both answers and outcomes data are scaled to the range $(0, 1)$.

## 3.1 A SYNTHETIC DOMAIN

**Task 1: A Simple Homogeneous Population.** In this task, each answer is independently drawn from a standard normal distribution and the outcome vector is constructed as follows: $y_{o,1} = y_{a,1}, y_{o,2} = y_{a,2}, y_{o,3} = 2\mu_1 - y_{a,1}$[4], and $y_{o,i} = 0, \forall i > 3$. Hence, only questions $\{1, 2\}$ are informative and only treatments $\{1, 2, 3\}$ should be considered. Prior to the survey phase, the decision maker does not know which treatment in $\{0, 1, 2\}$ is the optimal one since they have the same statistic. An optimal decision maker should only need to ask at most two questions (i.e. 0 and 1) to find the best treatment. Hence, we choose $Q_{\text{allow}} = 2$ here.

The result of this task is shown in Figure 3 (top row). Only LinUCB, HES, and RidgeUCB converge to zero per-period regret, and RidgeUCB takes the most number of users (nearly 60,000) to converge. HES can quickly find informative questions and well-targeted treatment as shown by the fact that its fraction of informative questions and treatment approaches 1 over time. RidgeUCB takes much longer to do so. When there are many uninformative questions, the uniform query strategy is not effective, as illustrated by the fact that UniformBOED, US, and BayesLinUCB do not converge. Since the answers are independent in this setting, inferring full context after uniformly querying questions is also not effective. In addition to HES, some of the other baselines such as US can achieve high accuracy in outcome prediction when conditioning on the observed answer. This high accuracy in outcome prediction does not necessarily imply high quality in decision making,

---

[3]For example, with the budget of $Q_{\text{allow}} = 3$, if RidgeUCB requests to observe question $\{1, 2, 4\}$, it is allowed to observe all three answers. Meanwhile, if it requests to observe $\{1, 2, 4, 6\}$, it will only observe one subset size 3 of its request, which could be $\{1, 2, 6\}$ or $\{1, 4, 6\}$

[4]$2\mu - x$ has the effect of geometrically reflecting a sample through the mean. For example, if the normal distribution is centered at 5, and $x = 4$, then $2\mu - x$ is 7. If $x = 8$, $2\mu - x$ is 2. Only 50% of the time $x$ would be higher $2\mu - x$ and is the better treatment.

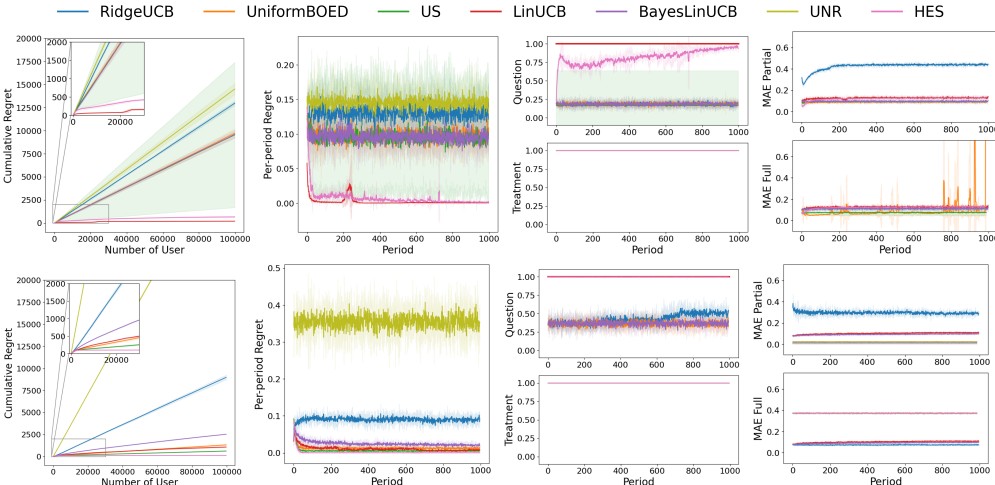

Figure 4: *A Homogeneous Population with Many Potential Treatments* (top) and a *Heterogeneous Population* (bottom). Summary statistics of all metrics are given in Table 3 and 4 in Appendix B.

illustrated by the fact that the per-period regret of UniformBOED does not converge to zero while LinUCB achieves sublinear regret even with relatively high MAE. In the domain where explainability is important, the action taken by HES can also be explained by its learned covariance matrix (Figure 7 in Appendix B). The fact that the learned covariance matrix from HES is very close to the one induced from the data generating process further confirms that the algorithm works as expected.

**Task 2: A Simple Homogeneous Population with Answer Dependency.** This experiment illustrates an intuitive concept that was introduced in Section 1: when there are $Q_i$ informative questions but they are dependent, an optimal algorithm may need to ask less than $Q_i$ questions. Task 1 is modified by letting $y_{a,1} = y_{a,2}$. Similar to the previous experiment, questions $\{1, 2\}$ are informative about the optimal treatment, but an optimal decision maker who knows one will immediately know the other. Hence, for this experiment we choose $Q_{\text{allow}} = 1$. Here, treatments $\{1, 2, 3\}$ should be considered, but two of them are equally good. Similar to the previous experiment, HES converges quickly in all metrics after a few thousand users (Figure 3, bottom row). Unlike the previous task, RidgeUCB can no longer converge to zero per-period regret, which can be explained by the bias from correlated omitted variables in ridge regression. The ability to handle dependencies between answers is crucial in the survey bandit setting, and hence this condition is included in all subsequent experiments. Even though the data generating process has introduced dependencies between outcomes (Figure 7 in Appendix B), HES still performs well here in practice.

**Task 3: A Simple Homogeneous Population with Many Potential Treatments.** The previous experiments show that HES works well when there are three treatments to consider. In practice, there can be potentially many treatments to consider. Here we show that our model is still effective when there are many potentially good choices for treatment. The outcome model of Task 2 is modified as follows: $y_{o,1} = y_{a,1}, y_{o,2} = y_{a,2}, y_{o,3} = y_{a,3}, y_{o,4} = 2\mu_1 - y_{a,1}, y_{o,5} = 2\mu_2 - y_{a,2}, y_{o,6} = 2\mu_3 - y_{a,3}$. The set of informative questions is $\{1, 2, 3\}$ but, for the same reason as Task 2, only two of them should be asked. Hence, we use $Q_{\text{allow}} = 2$. HES converges quickly in all metrics, demonstrating its ability to handle a large number of good treatments (Figure 4, top row).

**Task 4: A Heterogeneous Population.** Real-world populations of users may not be homogeneous. In other words, their distribution might not have simple geometry, such as being unimodal. This experiment demonstrates that HES is capable of handling this situation. The data is generated according to the following hierarchical process, where users may come from one of two groups:

- For users in the first group, the answer is drawn from an isotropic multivariate normal distribution with a mean vector of $[1, 2, ..., Q]$ and variance of 1. The outcomes are $y_{o,1} = y_{a,1}, y_{o,2} = y_{a,6}, y_{o,3} = 2\mu_1 - y_{a,1}$, and $y_{o,i} = 0, \forall i > 3$.

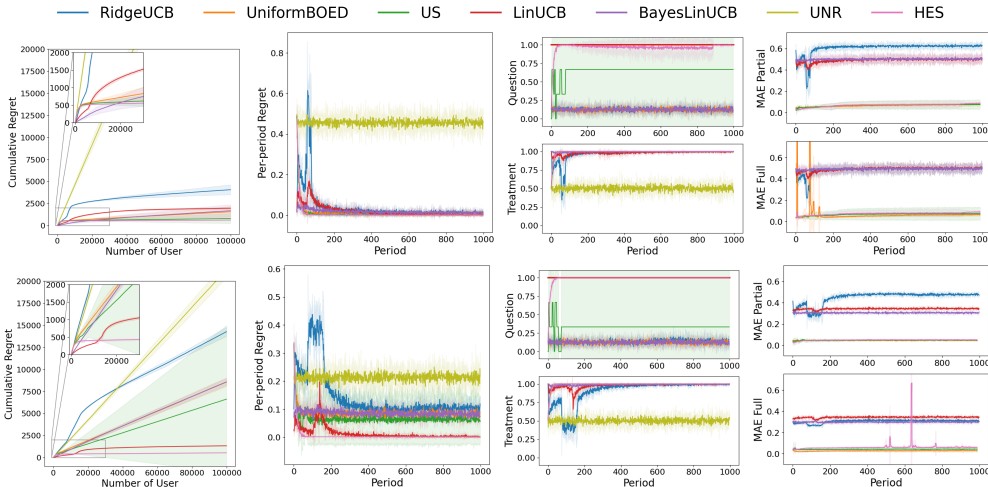

Figure 5: *A Skewed Population* (top) and *A Noisy Population* (bottom). Summary statistics of all metrics are given in Table 5 and 6 in Appendix B.

- For users in the second group, the answer is drawn from an isotropic multivariate normal distribution with a mean vector of $[Q, ..., 2, 1]$ and a variance of 1. The outcomes are $y_{o,1} = y_{a,2}, y_{o,2} = y_{a,7}, y_{o,3} = y_{a,12}, y_{o,4} = 2\mu_2 - y_{a,2}, y_{o,5} = 2\mu_7 - y_{a,7}, y_{o,6} = 2\mu_{12} - y_{a,12}$.

An optimal algorithm in this setting would need to ask four questions from the set $\{1, 6, 2, 7, 12\}$. Observing any first answer, the algorithm should know which cluster the user comes from[5], and asks at most 3 more questions to find the best treatments. Hence $Q_{\text{allow}} = 4$. HES converges quickly on all metrics, demonstrating its ability to handle the hierarchical data generating process (Figure 4, bottom row). Since there is no longer a simple linear relationship between the answer and outcome, even with context oracle, LinUCB does not converge to sublinear regret. This experiment highlights a strength of HES, that is with a flexible underlying probability model, it can handle several hierarchical, nonlinear data generating processes commonly found in practice.

**Task 5: A Skewed Population** As mentioned previously, the HES procedure for the survey bandit setting can be paired with different models. For demonstration purposes, in this paper, we use GMM, which strikes a good balance between expressiveness and statistical as well as computational efficiency. No matter the model, it is rarely a perfect fit for real-world data. In some modeling tasks, when the data distribution deviates from the assumed one, and if the underlying algorithm is not robust enough, the performance of the decision maker might degrade severely. This experiment shows that HES performance is robust to model misspecification. We modify Task 2 by generating answers from a skewed multivariate normal distribution, with skewness of 5, which we plot in Figure 8 (Appendix B); we note that this geometry is significantly different from the normal distribution. We choose the skewed multivariate normal family for this purpose because it allows us to control the dependency between variables while deviating from the typical Gaussian distribution. HES works well even in this case, showing that HES can achieve a certain level of robustness with distributional misspecification (Figure 5, top row).

**Task 6: A Noisy Population** It is common in practice to observe noisy measurements from each user. To examine the algorithm's robustness under noise, we add 10% normal noise to each observation (i.e. normally distributed noise with a standard deviation of 0.1 in our setting). Similar to the previous task, all metrics quickly converge for HES, demonstrating its robustness to noisy observations (Figure 5, bottom row).

---

[5]For example, when observing an answer to question 1, the algorithm should know the user belongs to group 1 if $y_{a,1} \approx 1$. Otherwise, $y_{a,1} \approx Q$ indicates that the user belongs to group 2

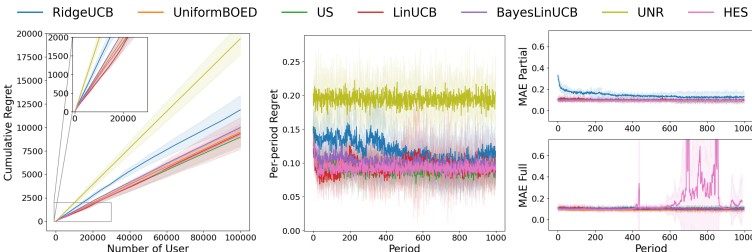

Figure 6: Results on the *Charitable Giving* experiment. Summary statistics for all of the metrics are given in Table 7 in Appendix B.

## 3.2 A Real Domain Based on Charitable Giving Experiment

In this section, a real dataset from a charitable giving experiment (Athey et al., 2022) is used to examine the algorithm in a practical setting. This dataset includes 3065 users and their answers to 16 general demographic characteristics, political affiliation, and media consumption. It also contains the response from each user to the following question: "How would people like you feel if we donated one thousand dollars to the following charity?" The user is then shown randomly 1 of 6 charities. Charitable giving is an especially suitable use case for adaptive experiments since the optimal policy is difficult to obtain beforehand. Behaviors that determine charitable giving are highly individual and depend on background and interest (Athey et al., 2022).

A semi-synthetic dataset that emulates the behavior of users from the above charitable experiment is generated using the following procedure. First, we learn a mapping from answers to outcomes. We train a number of models with 10-fold cross-validation on a 80:20 train:test split using MAE as the goodness of fit criteria. The following models are chosen: random forest (Athey et al., 2016), local linear forest (Friedberg et al., 2018), decision tree, a neural network with 1 layer of 17 hidden node and sigmoid activation, linear regression, Gaussian process regression, ridge regression, elastic net regression, Bayesian ridge regression, and a one-layer neural network with sigmoid activation. Hyperparameters of each model are tuned with a random grid search. The summary, given in Table 8 (Appendix B), indicates that the one-layer neural network works well for this dataset. This model is used to generate data for further experiments, where we draw context from a kernel density estimator (with parameters fit via 10-fold cross-validation) of the data, and map it to the outcome. We set $Q_{\text{allow}} = 2$ to emulate the time constraint when interacting with real users. The results of this experiment are shown in Figure 6. Here, HES achieves the same level of regret as LinUCB even with access to only 2 out of 16 questions. Compared to other algorithms, HES achieves a similar or better accuracy and regret level, demonstrating its usefulness in a real-world setting.

## 4 Discussion & Future Work

In this paper, we introduce a decision-theoretic entropy search procedure for the survey bandit setting, which treats the question phrase as a sequence of BOED tasks with the goal of asking informative questions to recommend treatments with minimal regret. The procedure may be paired with different models for answers and outcomes, and we demonstrate its usefulness on various synthetic and real datasets.

There are multiple future directions for this study. For example, some questions and treatments in practice are continuous (e.g. recommend 1mg dose or 5mg dose of some drug), which allows the EHIG optimization procedure to be done with gradient-based optimization. In addition, our $H_{r,\mathcal{A}}$-entropy search procedure is done myopically, which is computationally friendly but can be suboptimal. A non-myopic EHIG optimization procedure could achieve better performance. Third, we have considered the joint distribution of answer and outcome, but sometimes analyzing the semantic meaning of questions using tools from natural language processing can reveal information about the dependency of answers, even prior to observation of any answer. Last but not least, if computational speed is desirable, we want to explore the feasibility of a foundation decision maker that is pretrained using the EHIG objective, which might allow asking good questions in real time.

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

## A  GAUSSIAN MIXTURE MODEL DENSITY AND INITIALIZATION

This section includes details about the Gaussian mixture model (GMM), as well as the procedure for initializing the ECM algorithm under severe cases of missing data.

$$\Pr(c|y_{\text{obs}}) = \frac{p(y_{\text{obs}}|c)\Pr(c)}{\sum_{c'} p(y_{\text{obs}}|c')\Pr(c')} = \frac{\mathcal{N}(y_{\text{obs}}; \mu_{\text{obs}}^c, \Sigma_{\text{obs,obs}}^c)\Pr(c)}{\sum_{c'} \mathcal{N}(y_{\text{obs}}; \mu_{\text{obs}}^{c'}, \Sigma_{\text{obs,obs}}^{c'})\Pr(c')}$$

$$\mu_{\text{unobs|obs}}^c = \mu_{\text{unobs}}^c + \Sigma_{\text{unobs,obs}}^c \left(\Sigma_{\text{obs,obs}}^c\right)^{-1}\left(y_{\text{obs}} - \mu_{\text{obs}}^c\right) \tag{3}$$

$$\Sigma_{\text{unobs|obs}}^c = \Sigma_{\text{unobs,unobs}}^c - \Sigma_{\text{unobs,obs}}^c \left(\Sigma_{\text{obs,obs}}^c\right)^{-1}\Sigma_{\text{obs,unobs}}^c$$

When $C = 1$, GMM is a multivariate normal distribution:

$$y_o, y_a \sim \mathcal{N}(y_o, y_a; \mu, \Sigma); \quad y_a \sim \mathcal{N}(y_a; \mu_a, \Sigma_{aa}); \quad y_o \sim \mathcal{N}(y_o; \mu_o, \Sigma_{oo})$$

$$\mu = [\mu_a, \mu_o], \Sigma = \begin{bmatrix} \Sigma_{aa} & \Sigma_{oa} \\ \Sigma_{ao} & \Sigma_{oo} \end{bmatrix} \tag{4}$$

In the general case,

$$p(y) = \sum_{c=1}^{C} \Pr(c) \mathcal{N}(y; \mu^c, \Sigma^c); \quad \mu^c = [\mu_{\text{unobs}}^c, \mu_{\text{obs}}^c]; \quad \Sigma = \begin{bmatrix} \Sigma_{\text{unobs,unobs}} & \Sigma_{\text{unobs,obs}} \\ \Sigma_{\text{obs,unobs}} & \Sigma_{\text{obs,obs}} \end{bmatrix} \quad (5)$$

We propose the following initialization procedure to stabilize ECM optimization. This initialization is only done once for the entire interaction with all users. After the warm-up period, K-mean clustering is employed to assign the cluster membership to each data point. Next, within a cluster, for each outcome, a model that map answers to the outcome is learned, then used to infer unobserved outcomes. Note that this is only done for the purpose of initialization, and we do not alternate the data. Finally, the initial means and covariance matrices are initialized with the EM algorithm. From the second iteration, the initial mean and covariance matrix for each cluster is set to the ECM result of the previous iteration.

## B  EXPERIMENT DETAILS

This section contains further details on the empirical results.

| | Period | C. Regret | P. Regret | Questions | Treatments | MAE partial | MAE full |
|---|---|---|---|---|---|---|---|
| RidgeUCB | 50$k$ | $10978.36 \pm 107.43$ | $0.12 \pm 0.09$ | $0.46 \pm 0.38$ | $0.88 \pm 0.10$ | $0.41 \pm 0.05$ | $0.32 \pm 0.03$ |
| | 100$k$ | $11419.10 \pm 473.58$ | $0.00 \pm 0.00$ | $1.00 \pm 0.00$ | $1.00 \pm 0.00$ | $0.37 \pm 0.01$ | $0.37 \pm 0.01$ |
| UniformBOED | 50$k$ | $4922.05 \pm 160.79$ | $0.08 \pm 0.00$ | $0.13 \pm 0.02$ | $1.00 \pm 0.00$ | $0.04 \pm 0.00$ | $0.03 \pm 0.00$ |
| | 100$k$ | $9501.13 \pm 350.89$ | $0.10 \pm 0.01$ | $0.13 \pm 0.01$ | $1.00 \pm 0.00$ | $0.05 \pm 0.00$ | $0.03 \pm 0.00$ |
| US | 50$k$ | $3895.19 \pm 1965.96$ | $0.07 \pm 0.04$ | $0.17 \pm 0.24$ | $1.00 \pm 0.00$ | $0.05 \pm 0.00$ | $0.04 \pm 0.01$ |
| | 100$k$ | $7457.91 \pm 3962.57$ | $0.07 \pm 0.03$ | $0.17 \pm 0.24$ | $1.00 \pm 0.00$ | $0.05 \pm 0.00$ | $0.04 \pm 0.01$ |
| LinUCB | 50$k$ | $1187.45 \pm 45.94$ | $0.00 \pm 0.00$ | $1.00 \pm 0.00$ | $0.99 \pm 0.00$ | $0.36 \pm 0.01$ | $0.36 \pm 0.01$ |
| | 100$k$ | $1328.58 \pm 49.24$ | $0.00 \pm 0.00$ | $1.00 \pm 0.00$ | $1.00 \pm 0.00$ | $0.37 \pm 0.01$ | $0.37 \pm 0.01$ |
| BayesLinUCB | 50$k$ | $5061.14 \pm 184.07$ | $0.08 \pm 0.01$ | $0.15 \pm 0.04$ | $1.00 \pm 0.00$ | $0.31 \pm 0.01$ | $0.32 \pm 0.01$ |
| | 100$k$ | $9666.61 \pm 425.59$ | $0.08 \pm 0.01$ | $0.16 \pm 0.02$ | $1.00 \pm 0.00$ | $0.32 \pm 0.01$ | $0.32 \pm 0.01$ |
| UNR | 50$k$ | $11277.78 \pm 306.51$ | $0.19 \pm 0.01$ | N/A | $0.49 \pm 0.02$ | N/A | N/A |
| | 100$k$ | $22514.54 \pm 580.06$ | $0.23 \pm 0.02$ | N/A | $0.55 \pm 0.05$ | N/A | N/A |
| HES | 50$k$ | $420.84 \pm 14.90$ | $0.00 \pm 0.00$ | $0.96 \pm 0.01$ | $1.00 \pm 0.00$ | $0.06 \pm 0.00$ | $0.06 \pm 0.00$ |
| | 100$k$ | $443.46 \pm 14.48$ | $0.00 \pm 0.00$ | $0.94 \pm 0.00$ | $1.00 \pm 0.00$ | $0.06 \pm 0.00$ | $0.06 \pm 0.00$ |

Table 1: Performance of the seven comparison methods on Task 1, at time step 50,000 and at time step 100,000.

| | Period | C. Regret | P. Regret | Questions | Treatments | MAE partial | MAE full |
|---|---|---|---|---|---|---|---|
| RidgeUCB | 50$k$ | $9193.79 \pm 26.68$ | $0.10 \pm 0.01$ | $0.12 \pm 0.04$ | $0.98 \pm 0.02$ | $0.48 \pm 0.01$ | $0.31 \pm 0.00$ |
| | 100$k$ | $14138.40 \pm 121.28$ | $0.13 \pm 0.02$ | $0.14 \pm 0.04$ | $0.99 \pm 0.01$ | $0.46 \pm 0.01$ | $0.31 \pm 0.00$ |
| UniformBOED | 50$k$ | $4359.72 \pm 85.50$ | $0.08 \pm 0.02$ | $0.16 \pm 0.04$ | $1.00 \pm 0.00$ | $0.04 \pm 0.00$ | $0.03 \pm 0.00$ |
| | 100$k$ | $8394.27 \pm 187.30$ | $0.08 \pm 0.02$ | $0.11 \pm 0.01$ | $1.00 \pm 0.00$ | $0.05 \pm 0.01$ | $0.03 \pm 0.00$ |
| US | 50$k$ | $4800.55 \pm 160.81$ | $0.08 \pm 0.01$ | $0.00 \pm 0.00$ | $1.00 \pm 0.00$ | $0.05 \pm 0.00$ | $0.04 \pm 0.00$ |
| | 100$k$ | $9324.77 \pm 345.62$ | $0.08 \pm 0.00$ | $0.00 \pm 0.00$ | $1.00 \pm 0.00$ | $0.05 \pm 0.00$ | $0.04 \pm 0.00$ |
| LinUCB | 50$k$ | $1156.81 \pm 6.77$ | $0.00 \pm 0.00$ | $1.00 \pm 0.00$ | $1.00 \pm 0.00$ | $0.34 \pm 0.00$ | $0.34 \pm 0.00$ |
| | 100$k$ | $1288.91 \pm 8.98$ | $0.00 \pm 0.00$ | $1.00 \pm 0.00$ | $1.00 \pm 0.00$ | $0.35 \pm 0.01$ | $0.35 \pm 0.01$ |
| BayesLinUCB | 50$k$ | $4404.91 \pm 126.64$ | $0.07 \pm 0.02$ | $0.16 \pm 0.04$ | $1.00 \pm 0.00$ | $0.31 \pm 0.01$ | $0.30 \pm 0.01$ |
| | 100$k$ | $8510.54 \pm 269.46$ | $0.08 \pm 0.02$ | $0.16 \pm 0.02$ | $1.00 \pm 0.00$ | $0.31 \pm 0.01$ | $0.30 \pm 0.00$ |
| UNR | 50$k$ | $10652.95 \pm 168.83$ | $0.19 \pm 0.01$ | N/A | $0.51 \pm 0.03$ | N/A | N/A |
| | 100$k$ | $21265.43 \pm 273.16$ | $0.22 \pm 0.01$ | N/A | $0.48 \pm 0.02$ | N/A | N/A |
| HES | 50$k$ | $381.46 \pm 5.58$ | $0.00 \pm 0.00$ | $1.00 \pm 0.00$ | $1.00 \pm 0.00$ | $0.05 \pm 0.00$ | $0.06 \pm 0.01$ |
| | 100$k$ | $381.52 \pm 5.60$ | $0.00 \pm 0.00$ | $1.00 \pm 0.00$ | $1.00 \pm 0.00$ | $0.05 \pm 0.00$ | $0.06 \pm 0.00$ |

Table 2: Performance of the seven comparison methods on Task 2, at time step 50,000 and at time step 100,000.

| | Period | C. Regret | P. Regret | Questions | Treatments | MAE partial | MAE full |
|---|---|---|---|---|---|---|---|
| RidgeUCB | 50k | $6503.22 \pm 36.24$ | $0.13 \pm 0.02$ | $0.18 \pm 0.01$ | $1.00 \pm 0.00$ | $0.44 \pm 0.00$ | $0.11 \pm 0.00$ |
| | 100k | $12969.74 \pm 110.00$ | $0.14 \pm 0.01$ | $0.18 \pm 0.01$ | $1.00 \pm 0.00$ | $0.44 \pm 0.00$ | $0.12 \pm 0.00$ |
| UniformBOED | 50k | $4881.15 \pm 116.11$ | $0.09 \pm 0.01$ | $0.20 \pm 0.00$ | $1.00 \pm 0.00$ | $0.09 \pm 0.00$ | $0.07 \pm 0.01$ |
| | 100k | $9722.36 \pm 224.76$ | $0.10 \pm 0.01$ | $0.19 \pm 0.01$ | $1.00 \pm 0.00$ | $0.10 \pm 0.01$ | $1.86 \pm 2.52$ |
| US | 50k | $4815.47 \pm 1955.06$ | $0.09 \pm 0.03$ | $0.17 \pm 0.24$ | $1.00 \pm 0.00$ | $0.09 \pm 0.00$ | $0.08 \pm 0.02$ |
| | 100k | $9537.75 \pm 3937.19$ | $0.10 \pm 0.05$ | $0.17 \pm 0.24$ | $1.00 \pm 0.00$ | $0.10 \pm 0.01$ | $0.08 \pm 0.02$ |
| LinUCB | 50k | $161.43 \pm 11.18$ | $0.00 \pm 0.00$ | $1.00 \pm 0.00$ | $1.00 \pm 0.00$ | $0.13 \pm 0.00$ | $0.13 \pm 0.00$ |
| | 100k | $188.25 \pm 11.40$ | $0.00 \pm 0.00$ | $1.00 \pm 0.00$ | $1.00 \pm 0.00$ | $0.14 \pm 0.00$ | $0.14 \pm 0.00$ |
| BayesLinUCB | 50k | $4845.81 \pm 108.51$ | $0.09 \pm 0.00$ | $0.22 \pm 0.03$ | $1.00 \pm 0.00$ | $0.10 \pm 0.01$ | $0.11 \pm 0.01$ |
| | 100k | $9585.75 \pm 203.49$ | $0.09 \pm 0.01$ | $0.21 \pm 0.01$ | $1.00 \pm 0.00$ | $0.10 \pm 0.01$ | $0.12 \pm 0.01$ |
| UNR | 50k | $7283.02 \pm 147.13$ | $0.15 \pm 0.01$ | N/A | $1.00 \pm 0.00$ | N/A | N/A |
| | 100k | $14501.58 \pm 288.05$ | $0.15 \pm 0.01$ | N/A | $1.00 \pm 0.00$ | N/A | N/A |
| HES | 50k | $522.76 \pm 11.26$ | $0.00 \pm 0.00$ | $0.85 \pm 0.03$ | $1.00 \pm 0.00$ | $0.13 \pm 0.00$ | $0.13 \pm 0.00$ |
| | 100k | $647.54 \pm 39.10$ | $0.00 \pm 0.00$ | $0.94 \pm 0.02$ | $1.00 \pm 0.00$ | $0.13 \pm 0.00$ | $0.13 \pm 0.00$ |

Table 3: Performance of the seven comparison methods on Task 3, at time step 50,000 and at time step 100,000.

| | Period | C. Regret | P. Regret | Questions | Treatments | MAE partial | MAE full |
|---|---|---|---|---|---|---|---|
| RidgeUCB | 50k | $4512.49 \pm 56.48$ | $0.09 \pm 0.00$ | $0.44 \pm 0.03$ | $1.00 \pm 0.00$ | $0.28 \pm 0.01$ | $0.07 \pm 0.00$ |
| | 100k | $8970.97 \pm 123.73$ | $0.09 \pm 0.01$ | $0.55 \pm 0.03$ | $1.00 \pm 0.00$ | $0.29 \pm 0.01$ | $0.07 \pm 0.00$ |
| UniformBOED | 50k | $699.78 \pm 28.54$ | $0.01 \pm 0.00$ | $0.32 \pm 0.02$ | $1.00 \pm 0.00$ | $0.02 \pm 0.00$ | $0.37 \pm 0.00$ |
| | 100k | $1313.25 \pm 55.84$ | $0.02 \pm 0.00$ | $0.32 \pm 0.01$ | $1.00 \pm 0.00$ | $0.02 \pm 0.00$ | $0.37 \pm 0.00$ |
| US | 50k | $354.61 \pm 14.06$ | $0.01 \pm 0.00$ | $1.00 \pm 0.00$ | $1.00 \pm 0.00$ | $0.02 \pm 0.00$ | $0.37 \pm 0.00$ |
| | 100k | $616.99 \pm 23.70$ | $0.01 \pm 0.00$ | $1.00 \pm 0.00$ | $1.00 \pm 0.00$ | $0.02 \pm 0.00$ | $0.37 \pm 0.00$ |
| LinUCB | 50k | $683.51 \pm 10.13$ | $0.01 \pm 0.00$ | $1.00 \pm 0.00$ | $1.00 \pm 0.00$ | $0.10 \pm 0.00$ | $0.10 \pm 0.00$ |
| | 100k | $1071.22 \pm 27.24$ | $0.01 \pm 0.00$ | $1.00 \pm 0.00$ | $1.00 \pm 0.00$ | $0.11 \pm 0.00$ | $0.11 \pm 0.00$ |
| BayesLinUCB | 50k | $1426.23 \pm 13.73$ | $0.02 \pm 0.00$ | $0.36 \pm 0.05$ | $1.00 \pm 0.00$ | $0.10 \pm 0.00$ | $0.10 \pm 0.00$ |
| | 100k | $2523.30 \pm 13.31$ | $0.02 \pm 0.00$ | $0.38 \pm 0.01$ | $1.00 \pm 0.00$ | $0.10 \pm 0.00$ | $0.10 \pm 0.00$ |
| UNR | 50k | $17757.66 \pm 61.97$ | $0.34 \pm 0.01$ | N/A | $1.00 \pm 0.00$ | N/A | N/A |
| | 100k | $35368.73 \pm 175.97$ | $0.35 \pm 0.02$ | N/A | $1.00 \pm 0.00$ | N/A | N/A |
| HES | 50k | $103.03 \pm 2.06$ | $0.00 \pm 0.00$ | $1.00 \pm 0.00$ | $1.00 \pm 0.00$ | $0.01 \pm 0.00$ | $0.37 \pm 0.00$ |
| | 100k | $108.65 \pm 3.09$ | $0.00 \pm 0.00$ | $1.00 \pm 0.00$ | $1.00 \pm 0.00$ | $0.01 \pm 0.00$ | $0.37 \pm 0.00$ |

Table 4: Performance of the seven comparison methods on Task 4, at time step 50,000 and at time step 100,000.

| | Period | C. Regret | P. Regret | Questions | Treatments | MAE partial | MAE full |
|---|---|---|---|---|---|---|---|
| RidgeUCB | 50k | $3353.60 \pm 115.59$ | $0.01 \pm 0.01$ | $0.13 \pm 0.04$ | $1.00 \pm 0.00$ | $0.63 \pm 0.01$ | $0.51 \pm 0.02$ |
| | 100k | $4055.85 \pm 281.01$ | $0.02 \pm 0.01$ | $0.11 \pm 0.01$ | $1.00 \pm 0.00$ | $0.63 \pm 0.01$ | $0.50 \pm 0.04$ |
| UniformBOED | 50k | $1036.73 \pm 151.11$ | $0.01 \pm 0.00$ | $0.13 \pm 0.02$ | $1.00 \pm 0.00$ | $0.07 \pm 0.00$ | $0.06 \pm 0.00$ |
| | 100k | $1555.66 \pm 302.87$ | $0.01 \pm 0.00$ | $0.13 \pm 0.02$ | $1.00 \pm 0.00$ | $0.08 \pm 0.01$ | $0.06 \pm 0.01$ |
| US | 50k | $666.12 \pm 135.91$ | $0.00 \pm 0.00$ | $0.67 \pm 0.47$ | $1.00 \pm 0.00$ | $0.08 \pm 0.02$ | $0.07 \pm 0.03$ |
| | 100k | $787.97 \pm 279.39$ | $0.00 \pm 0.00$ | $0.67 \pm 0.47$ | $1.00 \pm 0.00$ | $0.08 \pm 0.02$ | $0.07 \pm 0.03$ |
| LinUCB | 50k | $1767.77 \pm 44.04$ | $0.01 \pm 0.00$ | $1.00 \pm 0.00$ | $0.99 \pm 0.00$ | $0.51 \pm 0.02$ | $0.51 \pm 0.02$ |
| | 100k | $1951.94 \pm 55.99$ | $0.00 \pm 0.00$ | $1.00 \pm 0.00$ | $1.00 \pm 0.00$ | $0.50 \pm 0.02$ | $0.50 \pm 0.02$ |
| BayesLinUCB | 50k | $1052.06 \pm 194.28$ | $0.01 \pm 0.01$ | $0.13 \pm 0.02$ | $1.00 \pm 0.00$ | $0.51 \pm 0.02$ | $0.50 \pm 0.02$ |
| | 100k | $1674.66 \pm 347.80$ | $0.01 \pm 0.00$ | $0.11 \pm 0.03$ | $1.00 \pm 0.00$ | $0.51 \pm 0.02$ | $0.50 \pm 0.04$ |
| UNR | 50k | $22788.37 \pm 686.00$ | $0.47 \pm 0.03$ | N/A | $0.51 \pm 0.04$ | N/A | N/A |
| | 100k | $45515.89 \pm 1403.41$ | $0.46 \pm 0.03$ | N/A | $0.47 \pm 0.01$ | N/A | N/A |
| HES | 50k | $572.48 \pm 12.05$ | $0.00 \pm 0.00$ | $0.96 \pm 0.06$ | $1.00 \pm 0.00$ | $0.08 \pm 0.02$ | $0.08 \pm 0.02$ |
| | 100k | $622.55 \pm 39.06$ | $0.00 \pm 0.00$ | $1.00 \pm 0.00$ | $1.00 \pm 0.00$ | $0.09 \pm 0.00$ | $0.09 \pm 0.01$ |

Table 5: Performance of the seven comparison methods on Task 5, at time step 50,000 and at time step 100,000.

| | Period | C. Regret | P. Regret | Questions | Treatments | MAE partial | MAE full |
|---|---|---|---|---|---|---|---|
| RidgeUCB | 50k | $9189.22 \pm 97.94$ | $0.10 \pm 0.01$ | $0.12 \pm 0.02$ | $0.98 \pm 0.01$ | $0.49 \pm 0.00$ | $0.32 \pm 0.00$ |
| | 100k | $14261.52 \pm 256.18$ | $0.12 \pm 0.01$ | $0.11 \pm 0.01$ | $1.00 \pm 0.00$ | $0.47 \pm 0.01$ | $0.30 \pm 0.00$ |
| UniformBOED | 50k | $4412.02 \pm 131.56$ | $0.09 \pm 0.01$ | $0.13 \pm 0.02$ | $1.00 \pm 0.00$ | $0.04 \pm 0.00$ | $0.03 \pm 0.00$ |
| | 100k | $8550.20 \pm 242.70$ | $0.09 \pm 0.00$ | $0.12 \pm 0.01$ | $1.00 \pm 0.00$ | $0.04 \pm 0.00$ | $0.03 \pm 0.00$ |
| US | 50k | $3437.06 \pm 1961.55$ | $0.07 \pm 0.05$ | $0.33 \pm 0.47$ | $1.00 \pm 0.00$ | $0.05 \pm 0.00$ | $0.04 \pm 0.01$ |
| | 100k | $6612.09 \pm 4158.32$ | $0.06 \pm 0.04$ | $0.33 \pm 0.47$ | $1.00 \pm 0.00$ | $0.05 \pm 0.00$ | $0.04 \pm 0.01$ |
| LinUCB | 50k | $1176.87 \pm 13.78$ | $0.00 \pm 0.00$ | $1.00 \pm 0.00$ | $1.00 \pm 0.00$ | $0.35 \pm 0.00$ | $0.35 \pm 0.00$ |
| | 100k | $1325.94 \pm 16.11$ | $0.00 \pm 0.00$ | $1.00 \pm 0.00$ | $1.00 \pm 0.00$ | $0.34 \pm 0.00$ | $0.34 \pm 0.00$ |
| BayesLinUCB | 50k | $4433.89 \pm 120.93$ | $0.09 \pm 0.02$ | $0.14 \pm 0.01$ | $1.00 \pm 0.00$ | $0.31 \pm 0.01$ | $0.30 \pm 0.01$ |
| | 100k | $8569.10 \pm 197.38$ | $0.07 \pm 0.01$ | $0.12 \pm 0.00$ | $0.99 \pm 0.01$ | $0.31 \pm 0.00$ | $0.30 \pm 0.00$ |
| UNR | 50k | $10649.84 \pm 206.16$ | $0.22 \pm 0.01$ | N/A | $0.47 \pm 0.07$ | N/A | N/A |
| | 100k | $21280.99 \pm 443.96$ | $0.22 \pm 0.03$ | N/A | $0.49 \pm 0.02$ | N/A | N/A |
| HES | 50k | $459.53 \pm 12.97$ | $0.00 \pm 0.00$ | $1.00 \pm 0.00$ | $1.00 \pm 0.00$ | $0.05 \pm 0.00$ | $0.08 \pm 0.03$ |
| | 100k | $530.65 \pm 12.94$ | $0.00 \pm 0.00$ | $1.00 \pm 0.00$ | $1.00 \pm 0.00$ | $0.05 \pm 0.00$ | $0.06 \pm 0.00$ |

Table 6: Performance of the seven comparison methods on Task 6, at time step 50,000 and at time step 100,000.

| | Period | C. Regret | P. Regret | Questions | Treatments | MAE partial | MAE full |
|---|---|---|---|---|---|---|---|
| RidgeUCB | $50k$ | $6513.33 \pm 256.42$ | $0.10 \pm 0.01$ | N/A | N/A | $0.14 \pm 0.01$ | $0.11 \pm 0.00$ |
| | $100k$ | $11857.16 \pm 759.71$ | $0.11 \pm 0.01$ | N/A | N/A | $0.13 \pm 0.02$ | $0.11 \pm 0.00$ |
| UniformBOED | $50k$ | $4868.65 \pm 119.91$ | $0.09 \pm 0.00$ | N/A | N/A | $0.09 \pm 0.00$ | $0.09 \pm 0.00$ |
| | $100k$ | $9512.92 \pm 299.32$ | $0.10 \pm 0.01$ | N/A | N/A | $0.10 \pm 0.00$ | $0.09 \pm 0.00$ |
| US | $50k$ | $4546.51 \pm 268.68$ | $0.09 \pm 0.01$ | N/A | N/A | $0.09 \pm 0.00$ | $0.11 \pm 0.01$ |
| | $100k$ | $8960.10 \pm 508.46$ | $0.09 \pm 0.01$ | N/A | N/A | $0.09 \pm 0.00$ | $0.11 \pm 0.00$ |
| LinUCB | $50k$ | $4557.71 \pm 291.95$ | $0.10 \pm 0.02$ | N/A | N/A | $0.10 \pm 0.01$ | $0.10 \pm 0.01$ |
| | $100k$ | $9346.26 \pm 845.33$ | $0.09 \pm 0.01$ | N/A | N/A | $0.10 \pm 0.01$ | $0.10 \pm 0.01$ |
| BayesLinUCB | $50k$ | $5152.81 \pm 108.11$ | $0.10 \pm 0.02$ | N/A | N/A | $0.10 \pm 0.00$ | $0.10 \pm 0.00$ |
| | $100k$ | $9975.07 \pm 112.65$ | $0.10 \pm 0.00$ | N/A | N/A | $0.10 \pm 0.01$ | $0.10 \pm 0.01$ |
| UNR | $50k$ | $9752.67 \pm 438.24$ | $0.20 \pm 0.02$ | N/A | N/A | N/A | N/A |
| | $100k$ | $19432.71 \pm 842.94$ | $0.21 \pm 0.03$ | N/A | N/A | N/A | N/A |
| HES | $50k$ | $4635.96 \pm 302.34$ | $0.09 \pm 0.01$ | N/A | N/A | $0.09 \pm 0.01$ | $0.11 \pm 0.01$ |
| | $100k$ | $9220.06 \pm 557.25$ | $0.09 \pm 0.01$ | N/A | N/A | $0.09 \pm 0.00$ | $0.18 \pm 0.11$ |

Table 7: Performance of the seven comparison methods on Task 7, at time step 50,000 and at time step 100,000.

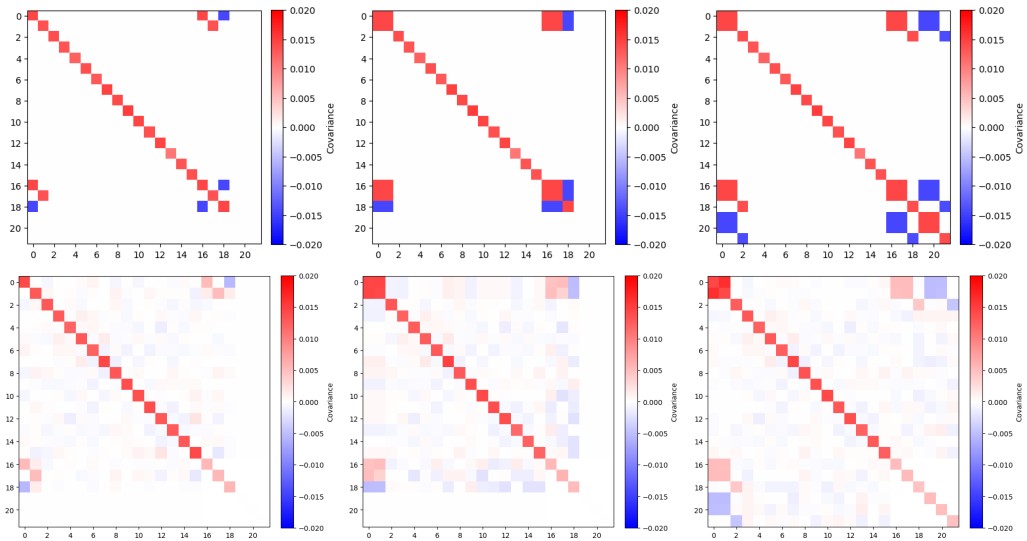

Figure 7: First line: Ground truth covariance matrix (estimated from the full population) in Tasks 1, 2, and 3 (from left to right). Second line: Recovered covariance matrix by HES in Tasks 1, 2, and 3 (from left to right).

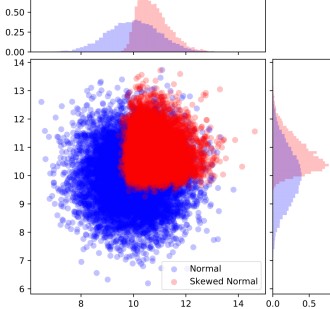

Figure 8: Skewed normal density

| | Local linear forest | Random forest | Linear regression | Ridge regression | Elastic Net regression | Decision Tree | MLP | GP |
|---|---|---|---|---|---|---|---|---|
| MSE | $0.23 \pm 0.01$ | $0.23 \pm 0.01$ | $0.24 \pm 0.01$ | $0.24 \pm 0.01$ | $0.25 \pm 0.01$ | $0.26 \pm 0.01$ | $0.23 \pm 0.01$ | $0.27 \pm 0.01$ |

Table 8: Performance of various learning algorithms on the *Charitable Giving* experiment.

