# OpenReview forum: "Bayesian Optimal Experimental Design for the Survey Bandit Setting"
_ICLR.cc/2023/Conference — Submitted to ICLR 2023_

### Official Review · Reviewer_5EZr · 2022-10-17

**Confidence:** 3
**Correctness:** 4
**Technical Novelty And Significance:** 2
**Empirical Novelty And Significance:** 3
**Recommendation:** 6

**Clarity, Quality, Novelty And Reproducibility:**

Quality:
The paper is generally high quality. The experiments section is well done.

Clarity:
The paper is very clear. The algorithm, choice of model and experimental settings are explained well.

Originality:
The paper is not very original. The setting was proposed by prior work, the entropy search method was obtained from (Neiswanger et. al., 2022) with minimal modifications, and the choice of model and optimization algorithm (ECM) are not novel either.

**Strength And Weaknesses:**

Strengths:
1. The setting is well motivated.
2. The algorithm is relatively straightforward.
3. The choice of GMM for modelling distributions over answers and outcomes is well justified.
4. The experiments section is excellent.

Weaknesses (more of clarifications):
1. Currently the algorithm is broken down into two parts, choosing a question with EHIG and then choosing a treatment with an $\epsilon$-greedy policy. Have the authors considered using the EHIG approach to choose both questions and treatments jointly? I imagine that this might lead to large cumulative regret if regret is measured based on the algorithm's chosen treatments, but perhaps this approach might be better if regret is measured based on the treatment with the highest posterior mean (i.e., immediate regret in (Hernandez-Lobato et. al., 2014)).

2. Does the LinUCB baseline take advantage of the distributions obtained from the GMM? If not, did the authors consider using the same $\epsilon$-greedy policy to choose treatments but with full access to answers instead? This seems like a more appropriate baseline to compare to see "what can be achieved when there is no budget constraint on question selection". I would expect such a baseline to always beat HES, while currently it seems HES beats LinUCB in some settings.

**Summary Of The Paper:**

The paper tackles the survey bandit setting in which the decision maker has a limited budget of questions to ask a user, observes the answers to these questions (constituting the features of the user), then recommends a treatment and observes the outcome. The questions to ask are chosen with a decision theoretic entropy search policy while the treatment is subsequently chosen with an $\epsilon$-greedy policy. The distributions over answers and outcomes is modeled with a GMM. The algorithm is compared to many baselines in different settings.

**Summary Of The Review:**

The paper is technically solid and provides an extensive empirical evaluation. However, it suffers somewhat from a lack of significant technical novelty as the core of the algorithm was obtained from (Neiswanger et. al., 2022) with minimal modifications required to make it work for the survey bandit setting.

---

### Official Review · Reviewer_XEp2 · 2022-10-24

**Confidence:** 3
**Correctness:** 2
**Technical Novelty And Significance:** 2
**Empirical Novelty And Significance:** 2
**Recommendation:** 3

**Clarity, Quality, Novelty And Reproducibility:**

I found the experiments section quite difficult to follow. In particular, this section raises the following questions which need to be addressed:

-Please provide a formal definition of regret. What is meant by the ‘optimal outcome’? Is the optimal outcome computed by conditioning on answers to all available questions? Or is this the optimal outcome computed by conditioning on answers to best Q_allow questions? In the latter case, is the optimal sequence adapted to the answers to the previous questions of a given user?

-Please provide a formal definition of the per-period accuracy. The optimal set of questions is context-dependent when given access to the answers to the already asked questions before choosing the next question. Are you comparing against the optimal batch of questions selected together at once per user to reveal the most informative contexts? If this is the case, evaluating the algorithm against this benchmark is not very informative, as the feedback available to the algorithm and the benchmark are different.

-How realistic are the experimental results given the length of the experiment (100.000 users) and initial warm-up (30 users per treatment)? While this many users may be typical for an online recommender system, it will be unrealistic to consider this many patient visits in healthcare.

-Synthetic domains in experiments consist of different tasks. In these domains, Q_allow is set based on the properties of the optimal decision-maker. This prior information is used by HES, which can provide a positive bias on the performance of HES. In the real world, Q_allow is determined by budget constraints. How will HES perform with respect to the other algorithms if Q_allow was set to a value different than what is set for the optimal decision maker?

-Please replace D with D_u when describing Equation 1.


**Strength And Weaknesses:**

Strengths:

-Survey bandit has interesting practical applications ranging from medicine to education.

-Decision-theoretic entropy search is an intuitive method for the sequential selection of questions.

-Proposed model can handle heterogeneous user populations and nonlinearities.

-Detailed experiments are performed. The proposed method is compared with several other benchmarks adapted to the survey bandit problem.

-Limitations are discussed.

Weaknesses:

-No theoretical analysis or convergence guarantees of the proposed method. The lack of theoretical results greatly hinders confidence in the algorithm. I think that theoretical regret analysis is an essential component (both lower and upper bounds on the regret) for a paper that uses Bayesian optimal experimental design techniques.

-Some of the performance metrics used in experiments are unclear. This raises several questions about experimental comparisons, which should be carefully addressed (please see comments in the next section).

-Scalability of the algorithm, both in terms of computing power and the number of required users, is unclear. Medicine can greatly benefit from sequential queries; however, convergence can still take a long time in terms of the number of users.


**Summary Of The Paper:**

This paper investigates a contextual bandit problem in which the decision-maker needs to actively acquire contexts in each decision epoch before the arm is selected. Ideally, the entire context vector should be revealed for optimal decision-making. However, budget constraints force the decision-maker to sequentially select a set of informative contexts to identify the optimal arm(s). Nevertheless, when context features are dependent, good decisions can be made even when a small set of contexts is observed. For efficient sequential query of the contexts, the paper proposes a decision-theoretic entropy search principle that identifies the context feature that provides the most information about the best action. After context acquisition is completed, arms can be selected by exploration-exploitation algorithms. The proposed acquisition is compared against several baseline methods via synthetic and semi-synthetic simulations.

**Summary Of The Review:**

This paper proposes an algorithm for the survey bandit problem. The problem itself has several potential practical applications. While the acquisition strategy is intuitive, theory and experiments do not fully reinforce the need for developing this new algorithm. A thorough theoretical analysis of the regret and experiments done using Q_allow different from the optimal Q_allow parameter will benefit the current work.

---

### Official Review · Reviewer_DagW · 2022-10-24

**Confidence:** 4
**Correctness:** 2
**Technical Novelty And Significance:** 3
**Empirical Novelty And Significance:** 3
**Recommendation:** 3

**Clarity, Quality, Novelty And Reproducibility:**

The paper is clearly written, even if it would require more formality in the problem definition and in the definition of the experimental metrics used. The setting analysed is not novel, but the techniques used by the authors are. The authors provided extensive details on the experimental setting. They also claim that the code will be released in the case the paper is accepted.

**Strength And Weaknesses:**

I think that the setting is interesting, but the work is not complete. It is true that from the experiments the authors provided that their approach seems to be effective, but no theoretical properties of convergence of the method are provided. It is common that, in practice, some heuristic methods are able to outperform the theoretically founded ones. However, without any theoretical results, we are not assured that the method will at least converge.

Moreover, there are still some questions about the experimental setting. For instance, the fact that the proposed method is able to outperform an oracle should be commented on and adequately motivated.


Minor comments:

"The decision maker’s goal ... in expectation." this is only a partial goal of the user. I think you should define more in detail this task and the overall goal.

In the experimental section, you should define the metrics in a formal way.

It is not clear to me how it is possible that the proposed method is able to outperform the LinUCB even if it has far more information than what has been proposed.

"During this period, ... one treatment per user." Is this phase required by all the methods? Why 30?

"Only LinUCB, HES, and RidgeUCB ... number of users." The comments in Figure 3 (top) do not correspond to the actual figure. I do not know which one of the two is the one to trust. The same holds for the bottom figure.

**Summary Of The Paper:**

The authors study the survey bandit setting, a scenario in which we are required to select sequentially the contextual information we want to be disclosed and, then, to select an option among a finite set. The authors used techniques from the Bayesian Optimal Experimental Design to build an algorithm for sequential selection of the context and of the option. The authors compared their approach with some baselines present in the literature and adapted for the setting.

**Summary Of The Review:**

The topic is suitable for the ICLR venue, but the proposed method requires a theoretical analysis before being published.

---

### Official Review · Reviewer_sGDT · 2022-10-28

**Confidence:** 4
**Correctness:** 3
**Technical Novelty And Significance:** 2
**Empirical Novelty And Significance:** 2
**Recommendation:** 3

**Clarity, Quality, Novelty And Reproducibility:**

**Clarity:**
The paper is well organized, but the presentation has minor details that could be improved, as discussed earlier in Strength And Weaknesses.

**Quality:**
The experimental evaluation is adequate and supports the main claims.

**Novelty:**
This paper contributes some new ideas, but they only represent incremental advances.

**Reproducibility:**
The code is unavailable, and some details about the experimental setup are missing, which makes it difficult to reproduce the main results.

**Details Of Ethics Concerns:**

I do not find any ethical concerns.

**Strength And Weaknesses:**

**Strengths of paper:**
1. The problem setting of *survey bandits* (gathering information about unknown contexts) is interesting. The proposed algorithm allows decision-makers to gather information about the unknown context by efficiently asking a sequence of questions.

2. The authors empirically validated the proposed algorithm's performance on synthetic and real datasets.

**Weakness of paper:**
1. Why not gather all possible information about context: Let's consider the treatment recommendation example used in the paper. Before recommending any treatment, the doctor must collect all possible information about the patient (except duplicate information). Because the cost of recommending the wrong treatment can be very high. Therefore, the authors must give a proper motivation for asking limited questions.

2. Finding the set of questions that gives maximum information about the user is an NP-hard problem (subset selection). However, selecting questions greedily one after another can be a good alternative, but its trade-off needs to be discussed.

3. Instead of using the decaying $\epsilon$-greedy strategy for action selection, authors can use the action strategies like upper confidence bound or Thompson sampling, which are much more efficient in dealing with exploration and exploitation.

4. Confusion about Line 1 after Section 2.1:  Assuming the known joint distribution (seems to be a typo in Line 1) of the answer and outcome can be impractical in many applications, and one has to estimate the distribution using observed data (Line 2). However, it seems that HES assumes the joint distribution is known (as doing the sampling in Line 7).

4. No theoretical guarantee: No theoretical guarantee of how the proposed algorithm will work compared to oracle policy (which selects the best questions in each round). Further, in the experiments, it is not clear whether the optimal treatment (action) uses all questions (full context) or the best subset of questions. Different regret notions need to be defined formally to avoid any confusion.

5. Minor issues in the algorithm: $\alpha$ is an algorithm input. It is unclear how the value of alpha is chosen and how it influences performance. For the first user, $D$ will be an empty set, so it is unclear how the $\theta_u$ will be computed.

**Summary Of The Paper:**

This paper studies a variant of the stochastic contextual bandits problem, where contextual information is not given but can be collected by asking a finite set of questions. The authors refer to this problem setting as *survey bandit*. The goal is to learn a policy that asks a series of questions that give the maximum information about the user (context) so that the policy can choose the best action for the given user and minimize the regret (the difference between the maximum achievable outcome and the policy's outcome).

The authors propose an algorithm named HES that uses the Bayesian optimal experimental design (BOED) approach to identify the next best question that gives the most information about the given user. They also validate the different performance aspects of the proposed algorithm on synthetic and real-world datasets.

**Summary Of The Review:**

This paper significantly overlaps with my current work, and I am very knowledgeable about most of the topics covered by the paper.

---

### Decision · Program_Chairs · 2023-01-20

**Decision:**

Reject

**Justification For Why Not Higher Score:**

This paper is preliminary: no regret bound and experiments raise questions.

**Justification For Why Not Lower Score:**

N/A

**Metareview: Summary, Strengths And Weaknesses:**

This paper studies a contextual bandit problem where the context is not perfectly known and can be obtained by asking a sequence of questions. While the setting is interesting and has several potential practical applications, it was proposed in another 2022 paper. The proposed algorithm does not come with a regret bound and several aspects of its empirical evaluation are unclear. The rebuttal was not submitted. This is likely because three reviews suggest a rejection.

I find it interesting that the authors study a graphical model problem. There are several recent papers that analyze bandit algorithms with graphical models, such as [Hierarchical Bayesian Bandits](https://proceedings.mlr.press/v151/hong22c.html) and [Metadata-based Multi-Task Bandits with Bayesian Hierarchical Models](https://proceedings.neurips.cc/paper/2021/hash/f7cfdde9db36af8e0d9a6d123d5c385e-Abstract.html). I hope that these papers can give the authors ideas for an analysis.